# Development of Digital Droplet PCR Targeting the Influenza H3N2 Oseltamivir-Resistant E119V Mutation and Its Performance through the Use of Reverse Genetics Mutants

**Laura A. E. Van Poelvoorde** [1,2,3,4], **François E. Dufrasne** [2], **Steven Van Gucht** [2], **Xavier Saelens** [3,4] **and Nancy H. C. Roosens** [1,*]

1   Transversal Activities in Applied Genomics, Sciensano, Juliette Wytsmanstraat 14, 1050 Brussels, Belgium
2   National Influenza Centre, Department of Infectious Diseases in Humans, Laboratory of Viral Diseases, Sciensano, Engelandstraat 642, 1180 Brussels, Belgium
3   Department of Biochemistry and Microbiology, Ghent University, 9052 Ghent, Belgium
4   VIB-UGent Center for Medical Biotechnology, VIB, 9052 Ghent, Belgium
*   Correspondence: nancy.roosens@sciensano.be

**Abstract:** The monitoring of antiviral-resistant influenza virus strains is important for public health given the availability and use of neuraminidase inhibitors and other antivirals to treat infected patients. Naturally occurring oseltamivir-resistant seasonal H3N2 influenza virus strains often carry a glutamate-to-valine substitution at position 119 in the neuraminidase (E119V-NA). Early detection of resistant influenza viruses is important for patient management and for the rapid containment of antiviral resistance. The neuraminidase inhibition assay allows the phenotypical identification of resistant strains; however, this test often has limited sensitivity with high variability depending on the virus strain, drugs and assays. Once a mutation such as E119V-NA is known, highly sensitive PCR-based genotypic assays can be used to identify the prevalence of such mutant influenza viruses in clinical samples. In this study, based on an existing reverse transcriptase real-time PCR (RT-qPCR) assay, we developed a reverse transcriptase droplet digital PCR assay (RT-ddPCR) to detect and quantify the frequency of the E119V-NA mutation. Furthermore, reverse genetics viruses carrying this mutation were created to test the performance of the RT-ddPCR assay and compare it to the standard phenotypic NA assay. We also discuss the advantage of using an RT-ddPCR instead of qPCR method in the context of viral diagnostics and surveillance.

**Keywords:** E119V mutation; influenza viruses; oseltamivir resistance; neuraminidase inhibition assay; RT-ddPCR





## 1. Introduction

Monitoring antiviral drug resistance in viruses and developing molecular methods for early diagnosis are considered to be of a priority level similar to that of the development of methods to detect bacterial strains with antimicrobial resistance [1–3]. Influenza A and B viruses are major causes of respiratory tract infections in humans. The hemagglutinin (HA) and neuraminidase (NA) glycoproteins are the main targets of the humoral immune response raised after natural infection or vaccination [4–6], whereas NA is a target of the main antiviral drugs licensed for the treatment of human influenza infection [7]. The major function of NA is to remove sialic acid residues that are present on newly synthesised virions and the membrane of infected cells. This activity of NA leads to the separation of the virus particle from the plasma membrane of the host cell and prevents self-aggregation and reattachment of the virus to infected cells [8,9] (Figure 1). Although vaccination remains the best way to prevent influenza, the limited coverage and moderate effectiveness of vaccination, the latter especially in the elderly, imply the need for efficient antiviral drugs as a complementary or alternative line of defence [10].

Currently, neuraminidase inhibitors (NAI), including oseltamivir and zanamivir, are the only effective antiviral drugs authorised in Europe for the treatment of influenza virus infection. Since their introduction in 1999, a few instances of naturally-occurring NAI resistance have been reported [7,11–13]. For example, during the 2007–2008 influenza season, oseltamivir-resistant H1N1 strains emerged and spread rapidly and globally. The tyrosine-to-histidine mutation in NA at residue 275 that confers resistance to NAI (H275Y-NA) was detected in almost all circulating H1N1 strains at that time. The H1N1pdm09 pandemic virus that emerged in 2009 was susceptible to oseltamivir and soon replaced the former H1N1. Nonetheless, for H1N1pdm09 viruses, substitution H275Y-NA is a major mutation conferring oseltamivir resistance [13–16]. In H3N2 viruses, a glutamate-to-valine mutation in NA at residue 119 (E119V-NA) and arginine-to-lysine mutation in NA at residue 292 (R292K-NA) are the major mutations conferring oseltamivir resistance [13–16]. These mutations interfere with the binding of oseltamivir to the active site of NA. At present, there is only a limited number of NAI-resistant influenza viruses circulating in the population and most cases are limited to immunocompromised patients undergoing treatment with NAI [17,18].

Surveillance systems have been implemented to monitor antiviral resistance to NAIs. The neuraminidase inhibition (NI) assay is the gold standard method to assess the susceptibility of an influenza virus to NAI. This assay allows researchers to determine the concentration of NAI needed to inhibit 50% of the activity of the NA enzyme ($IC_{50}$). The WHO Antiviral Working Group has defined three categories of NAI susceptibility: normal, reduced, and highly reduced inhibition [19]. A virus with reduced to highly reduced susceptibility to inhibition is considered drug-resistant [20]. However, the NI assay method requires virus propagation in cell culture, is labour-intensive, and relies on specialised laboratories [20–22]. Genotypic methods, such as PCR-based methods, have proven to be a valuable alternative for the rapid detection of known genetic modifications that confer antiviral resistance [23]. Indeed, for known mutations, they are more sensitive, less labour-intensive, and faster compared to phenotypic assays [1]. Therefore, multiple reverse transcriptase quantitative PCR (RT-qPCR) assays have already been developed to target the main mutations conferring antiviral resistance [1]. Their usefulness has previously been demonstrated for monitoring the emergence of resistant mutations in NAI-treated immunocompromised influenza patients, who often remain infected for a longer time than immune-competent patients [24].

More recently, ddPCR technology, which builds on established qPCR techniques, has become available [25]. This method is more commonly used than RT-qPCR because it has several advantages [26–29]. For example, when using the same primers and probe, a higher sensitivity and precision, as well as a lower susceptibility to PCR inhibitors [30], which could affect the Cq values and the accuracy of the viral quantification [31], have been reported for RT-ddPCR than reported for RT-qPCR [32,33]. In addition, RT-ddPCR gives an absolute quantification, which enables the comparison of different assays and laboratories without the need for a standard curve and calibration [33,34]. These advantages have led to an increasing number of clinical applications of ddPCR. It has been successfully applied in the certification of reference materials used for standardising qPCR assays widely employed in clinical diagnostics and research areas [32]. It allows accurate identification and quantification of viral DNA in clinical samples for diagnostic purposes. The fact that ddPCR is not prone to PCR inhibitors is also of interest when monitoring viruses in wastewater samples, as recently illustrated by the measurement of SARS-CoV-2 in wastewater [31,35]. Concerning the monitoring of major antiviral drug resistance, RT-ddPCR protocols have already been reported as having identified the H275Y-NA mutation in H1N1pdm09 viruses [36,37]. However, no ddPCR assay has been developed to detect substitutions E119V-NA and R292K-NA in H3N2 viruses.

In the present study, we propose several technical optimisations in terms of the annealing temperature and the choice of quencher for the development of a multiplex RT-ddPCR assay based on a previous qPCR assay developed by Van de Vries et al. (2013), which allows the detection and quantification of the E119V variant in influenza H3N2 samples [1]. To test the performance of the assay, a reverse genetics mutant carrying the E119V-NA mutation was generated. The specificity of the RT-ddPCR assay to detect a single mutation was successfully demonstrated. The sensitivity of the assay was tested using different defined proportions of the E119V-NA resistance mutation. Finally, the performance of the RT-ddPCR assay was compared to the phenotypic assay using reverse genetics viruses as the starting material.

## 2. Materials and Methods

### 2.1. Viruses, Cells, and Chemicals

A reverse genetics system available for wild-type Influenza A/Centre/1003/2012 (H3N2) in bi-directional pRF483 plasmids was provided by the Institut Pasteur, Paris, France. A coculture of Madin–Darby canine kidney (MDCK) and 293T cells was maintained in Dulbecco's modified Eagle's medium (DMEM) (Gibco) supplemented with 1% penicillin/streptomycin (Gibco). Oseltamivir carboxylate was obtained from Roche (Basel, Switzerland).

### 2.2. Mutagenesis and Reverse Genetics

Mutant NA segments were generated using the wild-type NA segment of the Influenza A/Centre/1003/2012 (H3N2) reverse genetics system using a QuickChange II Site-Directed Mutagenesis Kit (Agilent Technologies, Santa Clara, CA, USA) and GeneJET Plasmid Miniprep Kit (Thermo Fischer, Waltham, MA, USA) according to the manufacturer's instructions. The substitution E119V (GAA→G$\underline{T}$A) was introduced in the NA-plasmid (Figure 1). The sequences of the NA-plasmids were verified using Sanger sequencing and an Applied Biosystems Genetic Analyzer 3500 with the Big Dye Terminator Kit v3.1, following the manufacturer's instructions. Primers were used as described in Table S1. The results are shown in Figure S1.

All eight segments of the Influenza A/Centre/1003/2012 virus were used to generate the wild type and, for the mutant, the NA-plasmid was replaced by the mutant NA. Reverse genetics viruses were rescued by transfecting the MDCK/293T cells with eight plasmids corresponding to the eight virus segments using FuGene HD Transfection Reagent (Promega, Madison, WI, USA) and Opti-MEM (Gibco). The rescued virus was amplified via two passages on MDCK cells, and the supernatant containing the reverse genetics viruses was harvested by centrifuging for 5 min at $2000 \times g$ at 4 °C. The infectious virus titre was determined via a plaque assay as well as via qPCR [38].

### 2.3. Wild-Type and E119V Mutant Virus Mixes

The sensitivity of the ddPCR method was evaluated by making defined mixes of the wild-type virus and mutant virus of Influenza A/Centre/1003/2012 (H3N2). These mixes were made in triplicate in alignment with the PFU of the infectious wild-type virus and the mutant virus. The percentages of the mixes that were used are described in Table S2. For the NI assay, additional mixes were included in order to define the proportion of resistant virus at which the NI assay shows reduced inhibition (Table S2).



### 2.4. NA Activity and Inhibition Assays

The NA activity of the virus mixes was quantified with a fluorometric assay with the fluorogenic substrate 2′-(4-methylumbelliferyl)-a-D-N-acetylneuraminic acid (MUNANA) (100 μM). Based on the NA activity, each mix was diluted to the appropriate concentration with a dosage buffer (50 mL of MES, 20 mL of $CaCl_2$, and 430 mL distilled, sterilised water) and standardised to 10 μM 4-methylumbelliferone (MUSS). After an incubation period of one hour with the MUNANA substrate at 37 °C and NAIs present, a stop solution (50 mL glycine, 125 mL absolute ethanol, and 325 mL distilled, sterilised water) was added to measure the fluorescence. The NAI concentration (oseltamivir ranging from 0.05 nM to 5000 nM) that inhibited 50% of the NA activity ($IC_{50}$) was calculated using the variable-slope, four-parameter dose–response curve in GraphPad Prism 8. Antiviral susceptibility of influenza viruses is defined by criteria set by the WHO based on the fold change of the $IC_{50}$. For influenza A viruses, an $IC_{50}$ increase of less than 10-fold is categorised as normal inhibition. When a 10- to 100-fold increase in $IC_{50}$ is observed, it is categorised as reduced inhibition and, finally, if the fold change is more than 100-fold, it is categorised as highly reduced inhibition.

### 2.5. ddPCR Protocol

RNA was extracted from cell culture supernatants using the Easy Mag platform (BioMérieux , Marcy-l'Étoile, France) according to the manufacturer's instructions. An RT-ddPCR method for subtype A(H3N2) was developed using the primers and probes of the duplex qPCR method reported by Van de Vries et al. (2013). The method was also designed based on the manufacturer's instructions and the previously published ddPCR methods for the H275Y-NA mutation in H1N1pdm09 [36,37]. The digital droplet PCR was performed using the One-Step RT-ddPCR Advanced Kit for Probes (Bio-Rad) according to the manufacturer's instructions, with the exceptions that the mix was vortexed for 30 s at maximal speed after thawing and that, between every step, the mix was vortexed for 10 s at maximal speed. A total volume of 22 μL of mix was used for each reaction. The mix was set up on ice, including 0.99 μL of each primer (Figure 1) with an initial concentration of 20 μM (H3N2_E119V_FW_Primer 5′-AGGACAATTCGATTAGGCTTTCC-3′; H3N2_E119V_RV_Primer 5′-CTGTCCAAGGGCAAATTGAT-3′) and 0.55 μL of each probe with an initial concentration of 10 μM (H3N2_E119_probe FAM-ACA AGA GAA CCT TAT G-MGB-Eclipse; H3N2_V119_probe HEX-ACA AGA GTA CCT TAT G-MGB-Eclipse), 1.1 μL of 300 mM DTT, 8.12 μL of $dH_2O$, 2.2 μL reverse transcriptase, 5.5 μL One-Step Supermix, and 2 μL of the diluted sample (1:100). According to manufacturer's instructions, 20 μL of the sample mix and 70 μL of Droplet Generation Oil for Probes were loaded into a QX200™ droplet generator (Bio-Rad, Hercules, CA, USA). After the droplet generation, 40 μL of droplets was recovered per reaction. The RT-PCR amplification reaction was performed in a T100™ Thermal Cycler (Bio-Rad) with the following conditions: 1 cycle at 50 °C for 60 min (RT); 1 cycle at 95 °C for 15 min (Taq polymerase activation); 40 cycles at 95 °C for 30 s (denaturation) and at 55 °C for 60 s (annealing); and 1 cycle at 98 °C for 10 min (enzyme inactivation).The plate was then transferred to the QX200 reader (Bio-Rad) and results were acquired using the HEX and FAM channels, according to the manufacturer's instructions. For the interpretation of the results, the QuantaSoft software v1.7.4.0917 (Bio-Rad) was used and the threshold was set manually. For the ddPCR experiments, the average number of copies was calculated in MS Excel 2016. The false positive rate (FPR) was calculated by dividing the numbers of FAM-positive droplets in E119-NA controls and of HEX-positive droplets in V119-NA controls by the total number of positive droplets. The limit of detection (LOD) was calculated by adding three times the standard deviation to the mean of the FPR proportion or quantity.

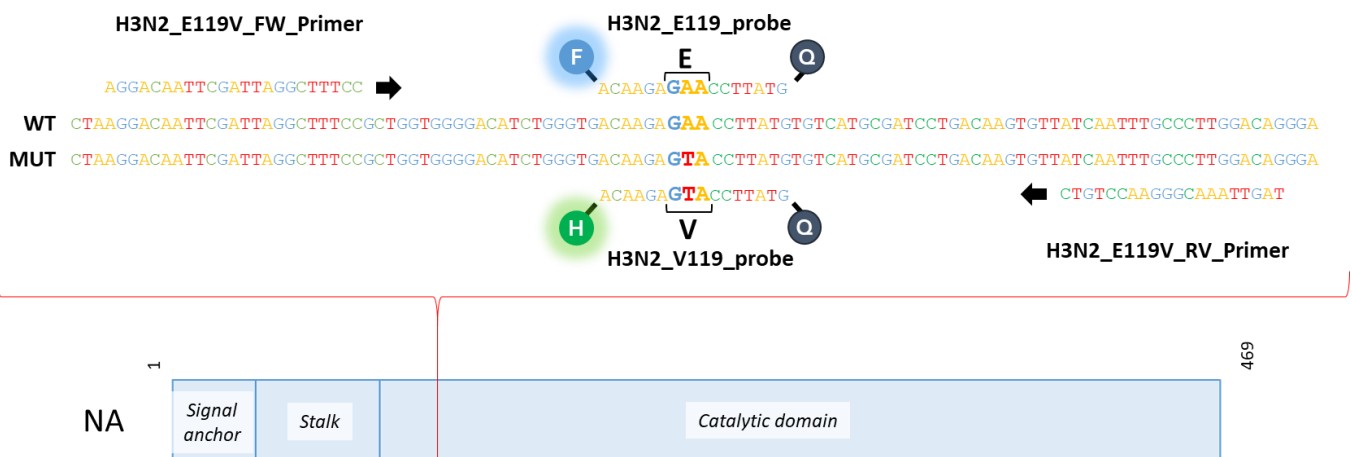

**Figure 1. Visualisation of NA segment with the position of the E119V mutation that is linked to antiviral resistance in A(H3N2) influenza viruses**. NA contains three major functional domains, namely the signal anchor that contains the transmembrane domain, and the stalk domain and catalytic domain [39]. The stalk domain is believed to position NA in respect to its receptor and the length varies within and across NA subtypes [40–46]. The catalytic domain is responsible for the viral release and it removes the cell surface receptor to prevent interparticle aggregation [47–52]. This contributes to the assembly process by tethering the stalk to the membrane in a tetrameric conformation. The modification introduced by reverse genetics allows a substitution of a glutamate-to-valine mutation in NA at residue 119 (red line). Therefore, the NA with E119V may accommodate a water molecule, which would interfere with the binding of oseltamivir to the active site. Moreover, the E119V mutation affected the optimal orientation of oseltamivir within the active site and also affected its overall conformation [11,53]. The primers and probes bind with the sequence and the two probes will allow the discrimination between glutamate (E, FAM in blue) and valine (V, HEX in green) at position 119.

## 3. Results

### 3.1. Generation of the Wild Type and E119V-NA Mutant

Viral titres of $1.53 \times 10^8$ and $1.50 \times 10^8$ were obtained for the wild type and E119V-NA mutant, respectively. Additionally, the RT-qPCR assay resulted in Ct values of 16.23 and 14.53 for the wild type and E119V-NA mutant, respectively.

### 3.2. Adaptation of RT-qPCR to Obtain a Specific RT-ddPCR to Detect the Polymorphism at Position 119-NA of Influenza A(H3N2) Viruses

The RT-ddPCR was developed based on a published RT-qPCR protocol [54]. Initial testing, using the primers and probes as described in this publication, on the RNA extracts of pure wild-type (E119-NA) or mutant (V119-NA) viruses showed poor discrimination of the positive and negative droplets in the fluorescent amplitude (Figure S2A,C). The change of the original BHQ-1 quenchers to MGB quenchers increased the specific binding of the probes and allowed for a better separation of the positive and negative droplets (Figure S2C,D). Using a thermal gradient, the annealing temperature was further optimised. Decreasing the annealing temperature from the original 62 °C [1] to 55 °C resulted in the best separation of the negative and positive droplets for both the wild-type and mutant viruses (Figure S3). After these optimisation steps, the four types of droplets could be sufficiently distinguished (Figure 2).

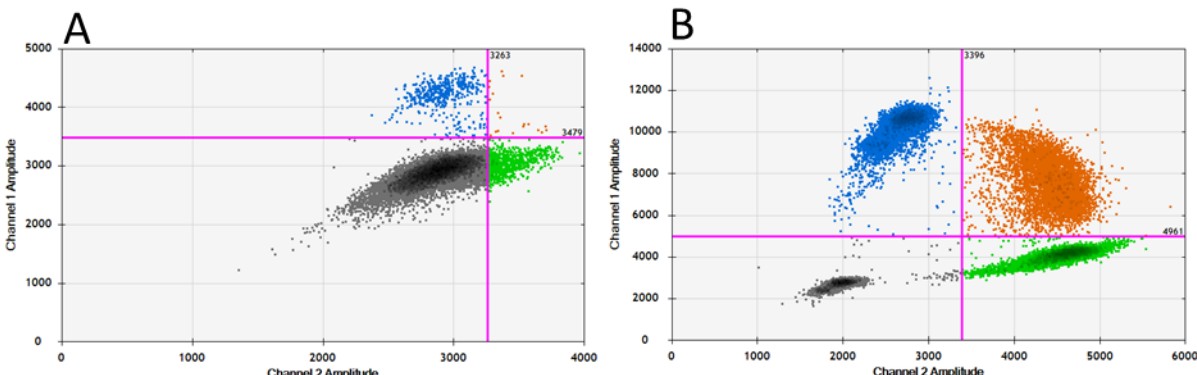

**Figure 2. QuantaSoft 2D view of results before (A) and after (B) optimisation of quenchers and thermal gradient.** The negative droplets are shown in black; droplets containing the FAM-labelled E119 wild type are blue and the HEX-labelled V119 mutant are green. Droplets containing both fluorescent probes are orange.

*3.3. Sensitivity of the RT-ddPCR to Quantify the Proportion of the E119V-NA Mutation*

Mixes of the wild-type and mutant viruses in a range of proportions were prepared in independent triplicates and, subsequently, the RNA extracts of these mixes were used to determine the limit of detection (Table S2). Figure 3 demonstrates that the goodness-of-fit measure is high ($R^2$ = 0.9959), with a slight underestimation of the proportion of the wild-type virus compared to the expected results. To evaluate the FPR, samples with only wild-type virus (WT100%–MUT0%) or only mutant virus (WT0%–MUT100%) were prepared in triplicate. The FPR is represented by the detection of V119-NA molecules in the wild-type virus, and E119-NA molecules in the mutant virus. The FPR was calculated by dividing the number of FAM-positive droplets by the total number of positive droplets in the mutant controls. The FPR was 0.21 ± 0.03% for the FAM probe (E119-NA) in the mutant controls (100% V119-NA) and 0.12 ± 0.08% for the HEX probe (V119-NA) in the wild-type controls (100% E119-NA). The limit of detection (LOD) of a method is associated with the FPR and its associated variance. The LOD of the RT-ddPCR protocol was calculated as the mean FPR proportion or quantity plus three times the standard deviation [37]. According to this formula, the LOD was set at 3.48 copies/µL, or 0.31%, for the FAM probe, and 1.91 copies/µL, or 0.37%, for the HEX probe.

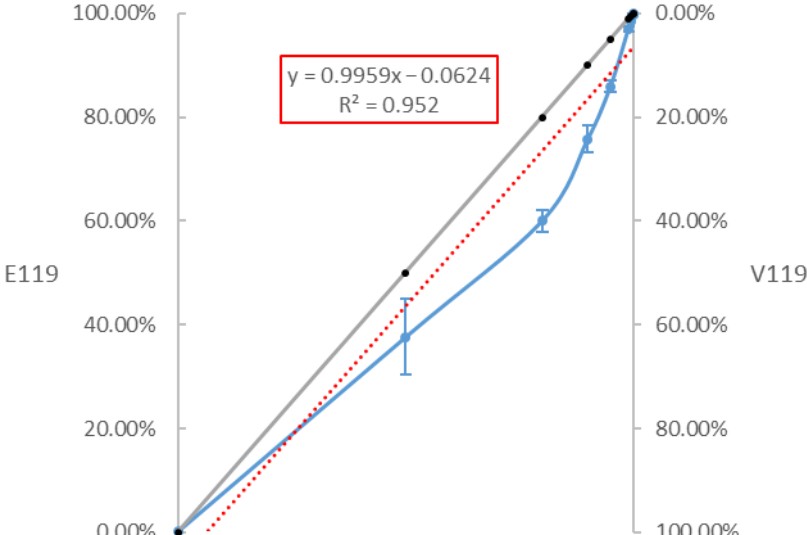

$$y = 0.9959x - 0.0624$$
$$R^2 = 0.952$$

**Figure 3. ddPCR results of the different proportions of the wild-type virus and the mutant virus.** The grey line represents the theoretical percentages of the mixes, while the blue line is the observed results. The red line represents the linear trend line of the obtained results.

*3.4. The Inhibitory Activity of Oseltamivir on Mixed Populations of NAI-Susceptible and -Resistant Influenza A(H3N2) Viruses*

First, the NA activity and NA inhibition assays were performed on the wild-type and E119V-NA mutant viruses. The E119V-NA mutant virus displayed a highly reduced inhibition phenotype (with an average $IC_{50}$ fold change of 177.6 $\pm$ 24.2 (CV = 13.62)) when compared to the E119-NA wild-type virus (Figure 4, Tables S2 and S3). Subsequently, these NA activity and inhibition assays were also performed on the 11 mixes (0.1%, 0.5%, 1%, 5%, 10%, 20%, 50%, 60%, 70%, 80%, and 90% E119V-NA mutant virus, respectively) of the wild-type and E119V-NA mutant viruses to identify the percentage of E119V mutant at which the NI assay can detect the presence of virus with the NAI resistance-associated mutation (Table S2). When the mix contained between 0.1% and 50% of the E119V-NA mutant virus, the $IC_{50}$ fold change remained below 10 and all the mixed viruses were considered as having normal inhibition of NI compared to the wild-type virus. When the mix comprised 60% E119V-NA mutant virus, the average $IC_{50}$ fold change was 11.6 $\pm$ 12.2 (CV = 105.27) (Figure 4, Tables S2 and S3). However, only one of the triplicates had a reduced inhibition phenotype with an IC50 fold change above 10 (Table S3). When the mix comprised 70% mutant virus, the three replicates showed reduced inhibition with an average IC50 fold change of 64.1 $\pm$ 35.1 (CV = 54.65) (Figure 3, Tables S2 and S3). Mixes of more than 80% mutant virus all displayed a highly reduced inhibition phenotype with IC50 fold-change values all above 100 (Tables S2 and S3).

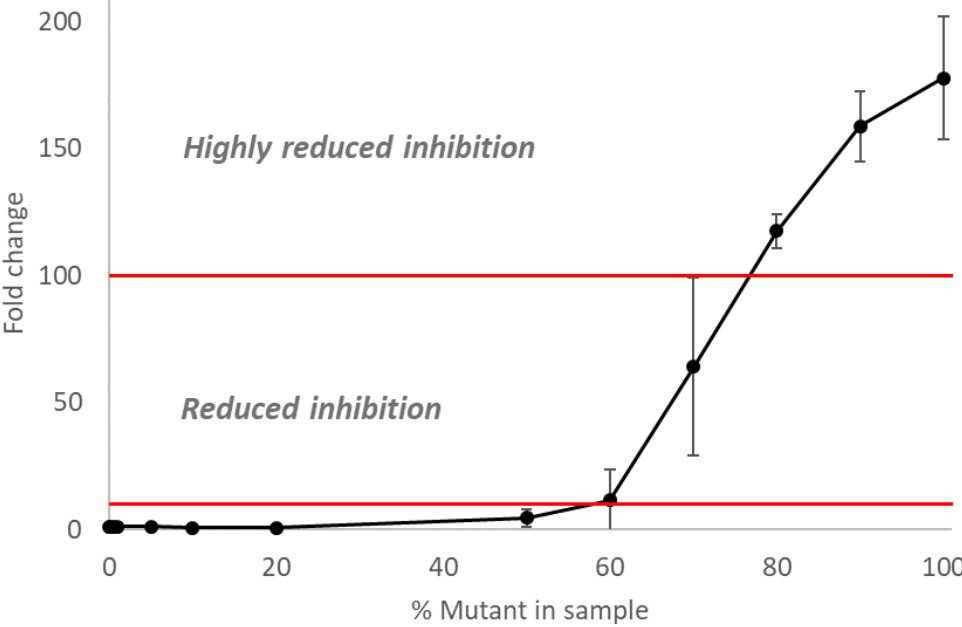

**Figure 4.** **Inhibitory activity of oseltamivir on mixed populations of NAI-susceptible and -resistant recombinant influenza H3N2 viruses with the E119V-NA mutation**. To compare the results, the fold change is used, which means that the IC50 of the tested virus is compared to the control. The fold changes at 10% and 100% are indicated by a red line. When the fold change is below 10, there is normal inhibition. When the fold change is between 10 and 100, there is reduced inhibition. If above 100, the fold change indicates highly reduced inhibition [20].

## 4. Discussion

The early detection of drug-resistant influenza viruses is important for patient management and for the prevention of the spreading and emergence of resistant strains in the population [55]. In the present study, an RT-ddPCR multiplex assay was successfully developed to detect the NAI-associated E119V mutant in influenza H3N2 viruses, which is one of the major mutations in H3N2 viruses conferring oseltamivir resistance. It is possible for the NA with E119V to accommodate a water molecule, which would interfere

with the binding of oseltamivir to the active site. Moreover, the E119V mutation affected the optimal orientation of oseltamivir within the active site and also affected its overall conformation [11,53].

To the best of our knowledge, no RT-ddPCR protocol has yet been described to detect and quantify this mutation. The performance of our ddPCR was evaluated to an LOD of 3.48 copies/µL, or 0.31%, for the E119 probe, and 1.91 copies/µL, or 0.37%, for the V119 probe. This result is in the same range as previously published RT-ddPCR assays. Indeed, for a first assay targeting the H275Y mutation in H1N1pdm09 viruses, an LOD of 0.28% oseltamivir-resistant influenza virus in a wild-type population and an LOD of 0.55% wild-type influenza virus in an oseltamivir-resistant population were observed [37]. In another study, the ability of the RT-ddPCR assay to detect rare SNPs was demonstrated up to 0.1% abundance [36]. When the current assay is compared to another qPCR method, the sensitivity of the former is higher. Indeed, the LOD of RT-qPCR using the same primers and probe was estimated in a previous study to be 5% in mixtures containing both wild-type and mutant viruses with approximated limits of detection of 2.4 $\log_{10}$ virus particles per millilitre [1].

The NI assay performed with the reverse genetics E119V-NA mutant indicates a detectable oseltamivir resistance phenotype when the proportion of E119V-NA mutant virus in the viral population is 60% or higher. This seems to be a higher proportion than reported in previous studies for H1N1pdm09 (25%) [56], and influenza B viruses (10% or 20%) [57]. However, the E119V-NA mutation can be observed with higher sensitivity using the RT-ddPCR method when compared to the NI assay and RT-qPCR method.

Our ddPCR method provides, therefore, an accurate and sensitive method to monitor drug resistance during treatment, which enables adaptation of the antiviral treatment over time. Moreover, the advantage of using RT-ddPCR instead of RT-qPCR is the possibility of absolute quantification of the viral RNA, which enables an easier comparison of different assays and laboratories without the need for a standard curve or calibration of reference material [33,34].

In addition to the diagnostics of clinical samples and the certification of reference material, ddPCR may prove to have potential uses for the surveillance in wastewater of specific mutations of pathogens present in the population. Indeed, compared to RT-qPCR, the RT-ddPCR method has the advantage of being less prone to PCR inhibitors. Furthermore, in the case of wastewater samples, the virus concentration will be relatively low; however, the RT-ddPCR method will enable the quantification of the virus and possible mutations at very low levels. This was recently illustrated in the context of the detection of variants during the COVID-19 pandemic [31,58]. As influenza can be detected in human excrement and wastewater [59], similar surveillance could be explored within a population to detect specific nucleotide variations in the circulating influenza viruses, including antiviral resistance mutations.

Finally, one of the factors that may hamper the development of a PCR method is a lack of materials, such as virus culture or clinical samples, to test its performance. Such materials may be challenging to obtain both in the case of new mutations and in general circumstances, when the mutation circulating in the population is low, which is currently the case for antiviral-resistant viruses. In the present study, a reverse genetics E119V-NA mutant virus was generated. This illustrates that reverse genetics methods that allow the mutation of specific nucleotides in the influenza viral genome can provide rapidly customised mutant viruses with the desired mutation [60] and may, therefore, represent a convenient alternative to clinical samples when the samples with the mutated virus are difficult to obtain. Moreover, reverse genetics viruses present the advantage of allowing the design of samples with appropriate mixes of the wild-type and mutant strains to assess the performance of the methods.

In conclusion, our developed multiplex RT-ddPCR method provides a sensitive and robust molecular assay to detect and quantify the E119V-NA mutation in H3N2 viruses. The RT-ddPCR method can detect the E119V-NA substitution with higher sensitivity than the NI assay when using reverse genetics viruses. After additional testing, this method may have, therefore, the potential to facilitate the calibration of reference material. This assay may also help support earlier detection of a virus in clinical samples and consequently the modification or change of antiviral treatment, resulting in an overall reduction in the use of ineffective drugs and, consequently, a reduction in costs and drug resistance. Moreover, the RT-ddPCR method may have the potential to enable the surveillance of antiviral resistance in a population using wastewater samples.

**Supplementary Materials:** The following supporting information can be downloaded at: https://www.mdpi.com/article/10.3390/cimb45030165/s1, Table S1: Primers used for Sanger sequencing to verify the sequences of the mutant A(H1N1)pmd09 and A(H3N2) NA-plasmids; Figure S1: Results of sequence of mutant and wild-type H3N2 around the mutation E119V; Table S2: Mean results for the RT-ddPCR and the phenotypic test; Table S3: Raw ddPCR data and phenotypic test data; Figure S2: QuantaSoft 1D view of results for optimisation quencher; Figure S3: QuantaSoft 1D view of results for optimisation of the annealing temperature using a thermal gradient.

**Author Contributions:** Conceptualisation, S.V.G., X.S. and N.H.C.R.; data curation, L.A.E.V.P.; formal analysis, L.A.E.V.P.; funding acquisition, N.H.C.R.; investigation, L.A.E.V.P.; methodology, L.A.E.V.P.; project administration, N.H.C.R.; supervision, F.E.D. and N.H.C.R.; validation, L.A.E.V.P.; visualisation, L.A.E.V.P.; writing—original draft, L.A.E.V.P.; writing—review and editing, F.E.D., S.V.G., X.S. and N.H.C.R. All authors have read and agreed to the published version of the manuscript.

**Funding:** This work was supported by the BE READY project funded by Sciensano.

**Institutional Review Board Statement:** Not applicable.

**Informed Consent Statement:** Not applicable.

**Data Availability Statement:** The data presented in this study are available within this article or in the Supplementary Material.

**Acknowledgments:** We would like to thank Cyril Barbezange for his support and expertise.

**Conflicts of Interest:** The authors declare no conflict of interest.

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
