# Peer review of "Development of Digital Droplet PCR Targeting the Influenza H3N2 Oseltamivir-Resistant E119V Mutation and Its Performance through the Use of Reverse Genetics Mutants"

_cimb, doi:10.3390/cimb45030165_

Round 1

Reviewer 1 Report

The manuscript entitled: ”Development of digital droplet PCR targeting influenza H3N2 oseltamivir-resistant E119V mutation and its performance through the use of reverse genetic mutants” is well written and addresses an important problem: antiviral drug resistance. Van Poelvoorde et al. focused their attention on the protein Neuraminidase (NA), as it is the main antiviral drug target. The team developed a reverse transcriptase droplet digital PCR assay.

 This article starts off by discussing the importance of monitoring antiviral drug resistance and the NA mutations leading to NAI resistance. In particular, the Authors paid attention to the E119V mutation. This is interesting because there is a loss of a negative charge. What does it mean? I think that a little investigation into this topic would be appreciated.

I have two major suggestions:

·      In the Introduction, the Authors could describe better the structure and the function of NA, also using a picture. I suggest inserting this new add-on after “human influenza infection [7]” (line 38).

·      In the Introduction, the Authors could investigate the impact of the E119V mutation in terms of negative charge loss. A picture showing electrostatic potential isocontours would be appreciated.

Please check the correct writing of the first author of references #1, #2, and #36

Reviewer 2 Report

Interesting and relevant topic is discussed.

Some revisions to manuscript are needed:

Section Introduction. Line 52- R292K-NA should be written first without abbreviations, as are in lines 15 and 46-47.

Section Materials and Methods. There is no information “wild -type virus”

Section Results. Lines 180-182 -The text is for Materials and Methods

Line 224 - It is written graph and table, but it is only a graph; There is no green line, there is gray line.

Section discussion. Lines 286-295- the information is unrelated to the topic of the article.

Round 2

Reviewer 1 Report

Dear Authors, in my opinion after the improvements the paper is ready to be published.